# IoT-Based Smart Water Management Systems for Residential Buildings in Saudi Arabia

**Rayed AlGhamdi** [1,*] **and Sunil Kumar Sharma** [2]

1 Department of Information Technology, Faculty of Computing and Information Technology, King Abdulaziz University, Jeddah 21589, Saudi Arabia
2 Department of Information Technology, College of Computer and Information Sciences, Majmaah University, Majmaah 11952, Saudi Arabia
* Correspondence: raalghamdi8@kau.edu.sa

**Abstract:** Water is a precious resource that can be intelligently managed. Effective water usage demands computerized home water supply management in a culture where water tanks, motors, and pumps are ubiquitous. Water management is crucial for the government and the citizens in countries like Saudi Arabia. The issue is providing a constant, high-quality, low-cost water supply. This study introduces a smart water management (IoT-SWM) system that may be used in structures that do not have access to a constant water supply but instead have water stored in enormous tanks underneath. The GSM module collects water use data from each home in a community and transmits it to the cloud, where it is analyzed. A smart water grid is a hybrid application that uses an inspection mode to identify leaks and measure the resulting height differences to keep track of the tank's water level. The system automatically deactivates the affected section after detecting any water shortage or malfunction in the system mechanism, such as broken valves, pumps, or pipes. It sends an emergency signal to building managers. It monitors essential water quality elements regularly, and if they fall below acceptable levels, it sends warning signals to the building management, who can take action. Over an extended period, the system monitored and recorded all water quality metrics. The system restarts when the water pump has been reconnected and sends an emergency alert. As a result, the suggested system has been an excellent replacement for Saudi Arabia's mechanically operated system.

**Keywords:** internet of things; smart water management; tank's water level; residential buildings





## 1. Introduction of Smart Water Management Systems

One of the essential elements in the universe is water. Nowadays, consumers continuously seek methods to simplify their lives [1]. Monitoring water quality is critical to ensuring the planet's health and long-term viability [2]. Water is the source of many infectious illnesses, and garbage thrown by residents and environmental disasters from industrial enterprises pollute most of the nearby freshwater supplies in SA [3]. Drinking water can be stored in an overhead tank [4]. The principal causes of water quality deterioration in residential buildings are the development of microbes in overhead tanks and distribution networks, corrosion of pipe material, and the non-replacement of existing pipes [5]. To avoid catastrophic health implications, it is necessary to continuously and remotely check the quality parameters of the water system in real-time [6].

Traditional water quality monitoring in South Africa (SA) is expensive and does not allow continuous and timely monitoring of water quality from various sources [7]. Sustainable water management strives to combine many water management areas and optimize advantages in SA [8,9]. This can be accomplished in various ways, including water reuse, collecting, and conservation techniques [10]. The delicate balance of nature is maintained through ecosystem processes, whereas human consumption leads to an

imbalance [11]. Sustainable water management can reduce water use by changing consumer habits and implementing water efficiency measures [12].

A key goal of sustainable and self-sufficient water management is to maximize water use at the regional or municipal level [13]. Information and control methods and monitoring leverage this resource [14]. Leakage can be reduced, quality assured, customer experience improved, and operations optimized through water management [15]. IoT can promote sustainable economic development and improved water resources and energy management in SA, which invests in citizens' well-being by supporting IoT adoption [16,17]. Additionally, water systems should be equipped with technology to create smart procedures [18]. In many water systems with weak infrastructure, uncertain supply, and customer satisfaction, or significant discrepancies between proportional bills and actual consumption or use, smart water systems can help improve the situation [19,20]. There are several benefits to adopting a smart water system, including minimizing financial losses, and creating new business models better to serve urban and rural populations [21,22]. The benefits of IoT technology in our smart water system project are well-known to us at this point. As a result, we will be able to control our energy consumption better and manage our resources. This project's primary objective is to design a novel, trustworthy, and adaptable water quality monitoring system for real-time monitoring of a remote water level throughout an IoT zone. Wireless sensor networks offer a novel framework for gathering and relaying data from various sources [23]. Extensive testing is performed on an Internet of Things system designed specifically for this network. The Internet of Things network's end goal is to allow for the monitoring and management of water supplies, distribution systems, and reservoirs. There has been extensive testing and analysis of this [24].

In this paper, hybrid applications and IoT devices are given prominence. Water is more commonly squandered at residences, and the major supply source is wasted. GPRS and GSM modules are the two IoT devices: a water tank level sensor that sends data to the cloud for analysis and a motor that turns on and off automatically. Using an IoT-SWM system, the water level can be monitored and controlled while leaks in the tank are detected and an estimated measurement is provided.

The main contributions of this paper are:

- Smart water management gives a greater understanding of the water system, including flaw detection, preservation, and water management.
- A comprehensive database of regions with water losses or unlawful connections can be built with the introduction of smart water system technology by public service corporations.
- Smart water grids can save costs by conserving water and energy while improving the quality of service to consumers. Wireless data transfer allows consumers to assess their water use to reduce water costs in other circumstances.

In this manner, the remaining components of the IoT-SWM system can be planned. Studies that are at the heart of this discussion are outlined in Section 2. The suggested study is summarized in Section 3, while the simulation results and comments are provided in Section 4. The report's final Section 5, delves deeply into the findings and progress.

## 2. Relevant Survey

Thermal characteristics have significantly impacted the environmental footprint and life-cycle evaluation of buildings. It is proposed in this study that the opacity of a façade element could be adjusted year-round in response to seasonal variations using the smart water-filled glass (SWFG) control technology described in this research. As a result of SWFG and WFG's climate-based design strategy, glass buildings are no longer seen as climate change liabilities and instead as opportunities for sustainability [25]. According to a new study, the Internet of Things (IoT) could monitor renters' and landlords' electricity and water use (IoT). One technique has made it simpler to read meters and water meters through the internet, and renters and landlords can see the data gathered through an

Android application [24]. Tenants and landlords have found the gadget and IoT-based Android app dependable, accurate, useful, and easy to use.

Domestic hot water (DHW) accounts for a significant portion of the energy consumption in modern homes. This study optimizes the input parameters and the neural network design to forecast residential hot water consumption [26,27]. DHW relies on graphical tools to analyze the predicted predictability of DHW consumption profiles for systems of various sizes and to illustrate the tight relationship between the predictability of a given profile and its coefficient of variation [27]. In the process sector, problem detection and diagnosis were widely used, and their applicability in building water supply systems is still completely untapped [28]. Water distribution system performance assessment rules (WDSPARs) were established for this study to detect frequent defects in the water distribution system of a building. The WDSPAR strategy for big non-residential building end-users had already been documented in a practical guide for the first time [29].

Underground water is increasingly depleted in many areas, putting present and future generations on the brink of being devoid of protection from rising climatic unpredictability. Therefore, information technology techniques and internet communication technologies (ICT) play a vital role in water resources management to minimize the excessive waste of fresh water and regulate and monitor water pollution [30]. ICT evaluated research that utilizes the Internet of Things as a communication tool that regulates the preservation of the available quantity of water, preventing waste by households and farmers [31]. It may serve as a paradigm for smart water management that uses the Internet of Things to separate monitoring and decision support from the coordination of business operations and subsystem implementation. The suggested approach for smart water management creates a unified environment in which equipment from different manufacturers may communicate and be managed effectively. Low efficiency in water distribution and consumption, system maintenance and improvement, and failure diagnosis can be traced back to the absence of standardization among water ICT equipment used by producers [32]. To better manage water distribution networks, the authors of this study combine the Internet of Things (IoT), Complex Event Processing (CEP), and declarative procedures to design an effective, efficient, and adaptable architecture they call REFlex Water. REFlex Water is the first solution to combine these technologies inside the framework of water delivery systems, according to the developers. This article provides an overview of the REFlex Water architecture and shows how it was applied to a functioning Brazilian municipal water system. After seeing positive results, Brazilian water business management sought to implement REFlex Water in different parts of the country's water distribution network [33]. This article describes how a smart water consumption measurement system was built, focusing on high decoupling, and integrating multiple technologies enabling the display of usage in real-time. A smart meter is a device that gathers information, which is then analyzed by a local server (Gateway) and sent to the cloud periodically. The leakage detection system then analyses the data, making the results available via a web interface. With a 100% accuracy, recall, precision, and F1 score for identifying leakage and a margin of error of just 4.63 percent, as evaluated by the water usage data, the algorithm surpasses the state-of-the-art alternatives [21].

The authors present a survey meant to sum up the current state of the art with regard to IoT-based smart water quality monitoring systems (IoT-WQMS), with a focus on those intended for use in the home, in light of recent developments in IoT that can be applied to the creation of more effective, secure, and inexpensive systems with real-time capabilities. This study looks at common WQM metrics, safe drinking water limits, and smart sensors. It validates contemporary IoT-WQMS with empirical measurements, discussion, and design recommendations. There is little question that this research will contribute to the growing industry of smart homes, workplaces, and cities [9]. To round off the setup, a Raspberry Pi microcontroller serves as the system's brain, analyzing and forwarding data from the sensor nodes to a web server. The central unit has access to online data via a monitoring system. Because of factors including sensor node size, power consumption, and regulatory

constraints, a 433 MHz connection between nodes was specified. It has been shown that the suggested design for sensor nodes is effective, with preliminary findings indicating that it may be used to monitor water and reduce losses [34].

The authors proposed an algorithm for automatic dynamic-cum-manual irrigation that considers each farmer's specific needs. The AgriSens' user interface is designed with farmers in mind, and it delivers data from the field to them via various channels, including a screen, a mobile phone, and a website. The data validation, delivery ratio, energy usage, and failure rate in varying climates and with variable irrigation treatments are all significantly improved. Greens have been shown in experiments to increase crop yields by at most 10.21% compared to conventional manual irrigation methods, to extend the network's lifetime by 2.5 times compared to the current system, and to maintain the dependability of 94% after 500 h of use [35].

The lack of access to reliable water sources is the most widespread and consequential of the many difficulties farmers face today. They suffer from a lack of water for farming because of the erratic supply of water caused by floods and droughts. Therefore, it is important to practice efficient water management and conserve systems so that water can be used effectively. This study offers an IoT-based water-management system to effectively use, preserve, and reuse water for crops, thereby resolving the issues farmers experience in the current state of water use in agriculture [36].

By contrast, while they use water, we studied certain investigations that protect water quality and decrease contamination. IoT-SWM system has been proposed to improve the stormwater quality, efficiency ratio, water demand ratio, leakage detection ratio, and non-revenue water ratio. The following section discusses the proposed IoT-SWM system briefly.

### 3. Proposed Method: IoT-Based Smart Water Management Systems

In Saudi Arabia, water supply and sanitation are marked by problems and successes. Water shortage is a major issue [30]. Water desalination, water distribution, sewerage, and wastewater treatment have received significant investments to combat water shortages [37]. Today, desalination provides roughly half of the country's drinking water, groundwater mining provides 40%, and surface water supplies in the mountainous southwest account for 10% [38]. A desalinated water pipeline runs from the Arabian Gulf to Riyadh, Saudi Arabia's capital and largest city. Consumers get water nearly at no cost. Despite these advances, the quality of service remains low, for instance, in terms of supply continuity. The public sector in Saudi Arabia is characterized by a widespread lack of institutional capability and governance. The Saudi vision 2030 addresses this challenge to be resolved [39]. Among the successes is expanding wastewater treatment and processed sewage to irrigate urban green spaces and farmland.

Figure 1 shows the smart water tank using IoT. A hybrid application and two devices make up the microgrid system. The first device measures the water tank's height and sends the real-time information to the cloud using a smart-level device. The GSM module of the smart level sends a signal to another device, a motor-controlled device, which automatically activates and deactivates the motor based on the signal. They activate and deactivate motors when they receive an input signal. With this technology, a leakage measurement hybrid application has been constructed. The device's ultrasonic Smart Level sensor continuously monitors the tank's height and uploads that data to the internet once per minute. With a microprocessor and UR detector, the GSM/GPRS module may send data to the cloud, where it can be stored and accessed remotely [40–42]. The effects are extraordinary. As the water level in the tank rises or falls, the intelligent level device sends a signal to the regulated motor device to turn the motor on or off, respectively. IoT devices upload information to the cloud, which can be evaluated later. Users can tell the system to alert them if a specific threshold is met. A system for intelligent water management should allow for constant monitoring of water levels. Overflows and leaks in water systems can be spotted quickly by real-time monitoring. They need a constant data connection and

a lot of juice to monitor in real time. Decisions can be made in real-time with the use of cloud computing. An increasing number of IoT devices are used in the water management system. Now that inexpensive sensors can be linked to the Internet of Things devices, we can more accurately assess water quality.

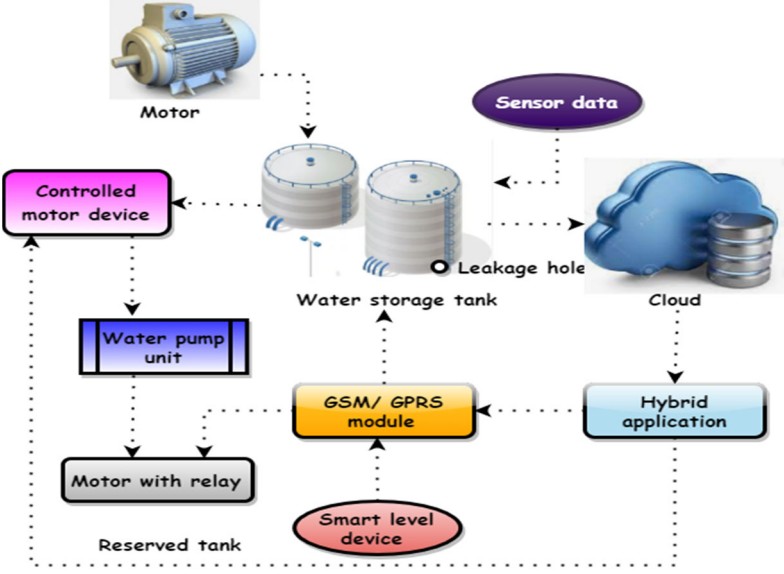

**Figure 1.** Smart water tank using IoT.

Relay-connected motors control the water pump, the second controlled motor device. The GSM module controls the motor and relay to respond to a signal and turn the motor on or off. A hybrid app must create both web-based and native hybrid applications. The cloud serves as a repository for the data collected, and it is this data that is analyzed and shown. The software will automatically check the water tank's current level since it regularly pulls data from the cloud. The application will signal the motor to turn the motor controller on or off. The app can remotely turn the motor on or off by sending a text message to the GSM network in the motor control device. If there is a leak in the water tank, the consumer can switch on the application's inspection mode at night to see any damage. Accordingly, it has been proposed as upcoming work to create a framework based on the Internet of Things for an efficient water management system that considers all these crucial characteristics and uses machine learning-based predictions to boost the smart management system's efficacy. Future works can additionally integrate the IoT coverage element while assessing measurement uncertainty.

Commercial time $w_k^t$ and water management fee $\rho$, as well as the link between the actual time $w_i^m$ and actual price $w_i^t$ is defined as

$$w_k^t = \rho \left( \sum_{i,m}^{N} G_{i,m} V_t u_i^m * e\left(w_i^m, w_i^t\right) \right) \tag{1}$$

As shown in Equation (1), traders $G_{i,m}$, current traditional finance $V_t$ claims are impacted by $u_i^m$ reporting, which they believe is illogical. The major service provider's profit function $U_z$ has not been affected by strategic forecasting $w$, which is given as,

$$U_z = w + \omega r_i - s_i + s(x - x_i)^2 \tag{2}$$

As shown in Equation (2), a major concern for bridge owners $\omega$ is the water transportation expenses $r_i$ since building resources $s_i$ and service provider's $s$ are often located near rivers $x$ and $x_i$ or the ocean.

After activating the mode, the consumer will not utilize the tank throughout the inspection. An application will record an inspection's beginning and ending times for use

in subsequent computations; if no leakage is detected, the inspection will end by itself after 6 h. The timer begins counting down as soon as the inspector mode is activated, and the application's leak detection and dimensioning calculations begin running in the background. A tank with a leak can be detected by comparing its current level to its initial value every 30 s and then comparing that value to the previous value retrieved from the cloud if there are any changes.

Figure 2 shows the IoT-based smart water management systems. Three-phase pumps, as previously indicated, are often used in water management systems for high-rise structures. This can lead to tanks being overfilled or pumps being overworked, which wastes water and energy and shortens the life of the pumps. For this reason, an intelligent water management system was created that can be used alone or as part of a larger building management system (BMS). Water level sensors in different tanks were linked to a direct digital controller (DDC), which controlled the whole system. The DDC controls the pumps through the pump panel to which it is connected. People can remotely operate the pump and check tank levels using a smart app with a DDC. It needs to be linked to a local Wi-Fi network or the internet to attain this purpose.

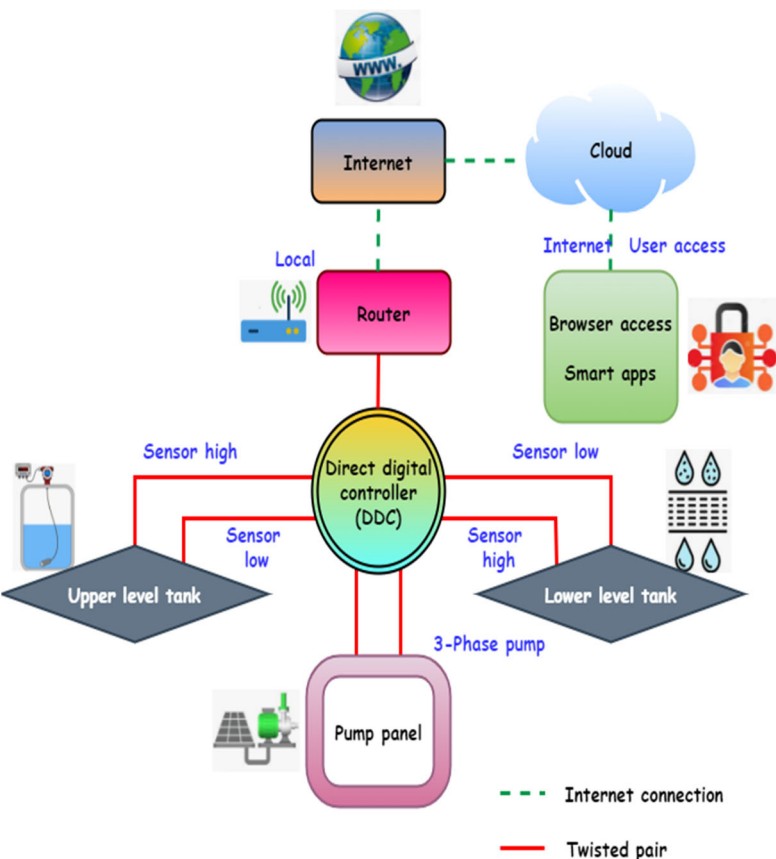

**Figure 2.** IoT-based smart water management systems.

The DDC can be linked to a router and used as a local Wi-Fi network or connected to the internet. In the case of a local network, people can use a smart app or a web browser on a connected device to access it directly through that network. The DDC is linked to the cloud server using the internet connection option. People get additional functionality when using the cloud server option by storing data on a remote cloud server. People can use the smart app or browser to send DDC instructions or get status information by establishing a cloud server connection. It is possible to keep the cloud server's pump and water level history if needed. An efficient water management system is essential for overcoming the difficulties associated with water shortages. Management of water resources is made feasible via continuous monitoring of quantity and quality. Monitoring water levels in

real time may drastically reduce water waste caused by overflowing storage tanks. By comparing the water levels at various times of day, the water management system may aid in detecting water leaks in a smart house.

Even though recycled water $x$ cannot seem appetizing, it is harmless and tastes just like conventional tap or mineral water $p$, which is given as:

$$p = \left( x + z^2 \right) \exp \omega^2 \sum w(k) \sqrt{m} \tag{3}$$

As shown in Equation (3), where the total amount of storage reservoirs is $z$, preparing $m$ and implementing a storage schedule is $w(k)$, and the exponential growth of the waste disposal site is $\omega$.

The factors of building time and cost can be used to measure the overall plan risk $X^{(i-1)}$, described as

$$X^{(i-1)} = H \frac{X^{(i)}}{\left| X^{(i)} \right|} - T_d \tag{4}$$

As shown in Equation (4), using land trusts $H$, property owners can retain ownership of their land while renting $X^{(i)}$ out to long-term infrastructure $T_d$ Initiatives.

The information and current condition can be accessed from Saudi Arabia over the internet. The cloud server provides the same if numerous locations must be collectively managed from a single centralized interface. Whether on their own or integrated into a building management system (BMS), water management systems can save money and energy by reducing water waste and ensuring that the energy company's pumps' energy consumption and pump life is extended. On-site worker productivity is increased using this method.

Figure 3 shows the strategy of the water supply network. During functional analysis, the activities of a system are represented by two types of functions: main and technical. The essential aim of a system's behavior is communicated by its functions, and the system's response to stressors imposed by the external environment is modeled using technical functions. Using functional block diagrams (FBDs), the system and its surrounding environment are represented in functional analysis. An external functional analysis aims to identify the system under consideration, its boundaries, and the external contexts with which it interacts. The urban system is defined by its physical and administrative limits when applied to Saudi Arabia. Using this FBD, people can see how the urban system interacts with its surroundings, including other cities and rural areas, as well as external technical networks (such as those for power and water, telecommunications, and transportation), as well as environmental factors (such as weather and seismic conditions).

Various urban engineering professionals gathered for brainstorming meetings to develop these concepts. An urban system can be distilled down to its most basic elements with this FBD. A typical urban system is used to conduct the internal functional analysis. Technical networks, housing, companies, and public infrastructure are among the many subsystems that urban planning experts say should be distinguished. Each of these four major subsystem groups includes the Saudi Arabian people. The FBDs have taken over most of the more specialized responsibilities. This is a good illustration of an FBD for a drinking water supply (DWS) technical network.

It does the same thing as a bank account trust fund. A trust bank $gt_1$ is a good option for those who cannot or do not want to maximize the use of their assets $s$ on their own and do not have the knowledge $\delta x_l$ or expertise to do it themselves, stated as,

$$\delta x_l = \frac{gt_1}{s(x_2 - x_1)(1 + t_1)} \tag{5}$$

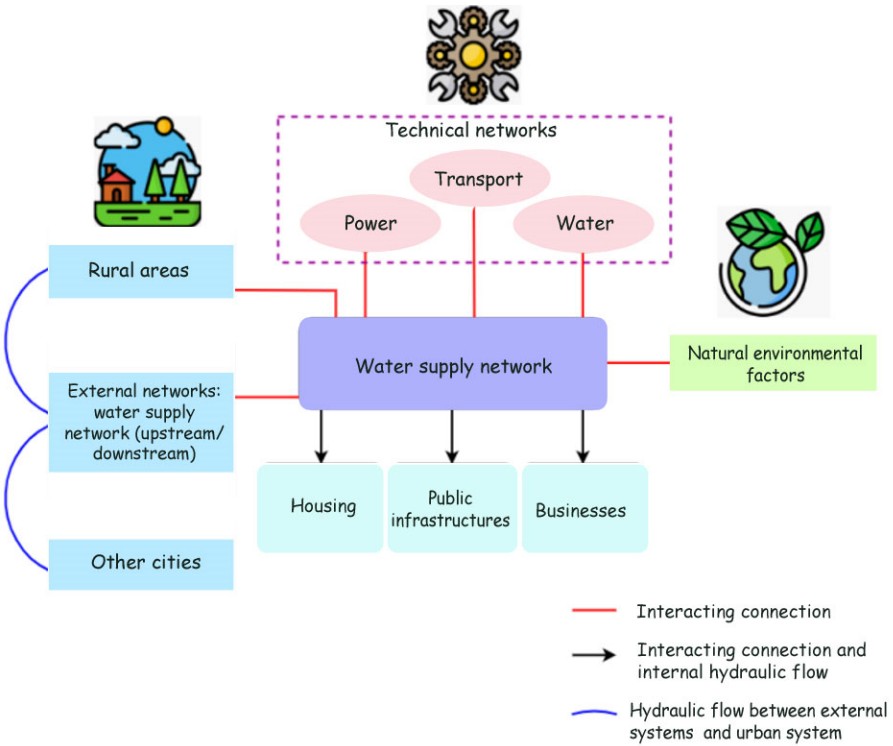

**Figure 3.** The strategy of a water supply network.

As shown in Equation (5), it is the owner's responsibility to secure the things $x_2 - x_1$ by making an acceptable reservation $t_1$.

The purpose of an index is to make complex statistics $k$ on water quality $W$ understandable to the general audience stated as,

$$|W| = R(k)^2 * \left(\prod \mu\right)\tau(k)\rho(k)R \int k \tag{6}$$

As shown in Equation (6), the pace at which data is moving $R(k)$, water that acts as a bridge $\mu$ between the earth's surface $\tau(k)$ and the atmosphere $\rho(k)$, and electrical conductivity temperature $R$ as a function of arithmetic.

Simple random $M$ is employed to ensure that a sample group $h$ of all responders is determined as,

$$M = \frac{Z^2 \times P + (P-1)}{h^2} \tag{7}$$

As shown in Equation (7), a different number of people $Z$ are randomly selected to represent the $P$ corresponding sample sizes.

To meet the facility's asset allocation $\rho(x)$, the building must be executed on schedule $DF_k$ and on budget $FG_k$, defined as

$$\rho(x) = \sum_{t=FG_k}^{DF_k} t.z_{kr} - q_{kr} \tag{8}$$

As shown in Equation (8), a company's average tenure $z_{kr}$, the work's initial and furthest completion dates, $q_{kr}$ and the number of $t$ direct predecessors.

Using a DWS technical network and an FBD, it is possible to distinguish and emphasize several main and technical roles. Drinking water for homes, companies, and public facilities and protection against fire in the event of a disaster are the primary purposes of this system. The technical functions of the DWS technical network include resistance to mechanical loads and resistance to failures in other technical networks. In a functional analysis table,

all of the findings of the internal functional analysis are summarized, outlining the primary and technical roles of each of a city's subsystems.

Figure 4 shows the drinking water distribution for residential buildings. There are two types of water in the reference system (RS), billed water (BW) and non-revenue water (NRW), as well as controlled and uncontrolled consumption. People pay directly for the water they use, referred to as billed water. The NRW is the total amount of water lost and consumed by authorized agencies, including the NRW. Due to natural losses, Saudi Arabia's water companies are judged on their administration and operations quality. The visible or monetary losses are due to unlawful use for water loss. The actual losses are water loss, bursting or explosion of pipes and reservoirs, and reservoir evaporation. Since it is impossible to distinguish between the apparent and actual amount of water lost, it is necessary to enhance their values by utilizing recorded readings and improving network monitoring (i.e., water metering).

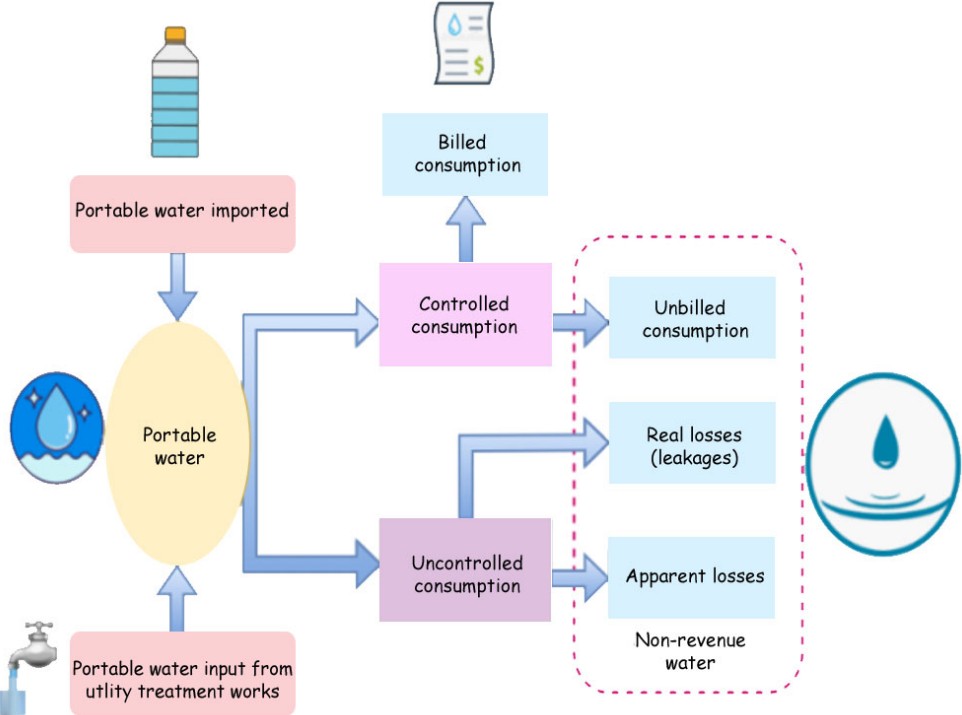

**Figure 4.** Drinking water distribution for residential buildings.

According to this suggested technique, Saudi Arabia can achieve the economic level of leakage (ELL) in water loss reduction. ELL represents the objective value of water companies, which involves looking for the highest amount of money that can be invested relative to the amount of water wasted. According to the analyzed case studies (RS and CMC), lowering water losses in the water distribution system is the best way to increase NRW. RS must have been imperative to enhance its water loss monitoring and management.

During the first stage of wastewater treatment $T$, solids are separated from wastewater and left to settle, defined as,

$$T = \varphi(r)[p]^2 \sqrt{\beta} \prod r \sum_k q \qquad (9)$$

As shown in Equation (9), data collection $\varphi$ and analysis at a water treatment facility $r$, some developers $k$ are working with pipelines $p$ that are largely intact and quality $\beta$ and efficiency of industrial water supply $r$, handling of industrial wastage $q$.

It is recommended to fix this mistake by using a cost index $Ek_{t,0}$ to provide pricing between two succeeding periods, described as,

$$Ek_{t,0} = \prod_{r=1}^{t} J_{r,r-1} - P_r \tag{10}$$

As shown in Equation (10), an arbitrarily selected baseline period $t$ and multiplied by the total of all successive cost gauges $J_{r,r-1}$, indicating the best method for $P_r$ computing a cost index.

Choosing suppliers based on the minimum bid price $N_k$ is the most common practice, given as,

$$N_k = \sum_{k=1}^{m} \frac{z_{ki} w_i}{\sum_{i=1}^{r} z_{ki}} \tag{11}$$

As shown in Equation (11), the relative relevance of sub-criteria, $z_{ki}$ signifies the service provider's evaluation based on the $w_i$ criterion, and indicates a large number of criteria.

This led to RS's primary monitoring tools ensuring that the system was running well in quantity and quality. Other advantages include identifying district metering areas' (DMA) consumption, irregular nocturnal use, and controlling water distribution network pressure. These devices made it possible to obtain data on water use, pressure variations, and flow or quality probes straight from the network. A central database stores all of the data acquired by these devices, decreasing the need for estimates by providing scans and regular and trustworthy recordings to the management organization.

Figure 5 shows the greywater recycling system for the residence. The concepts mentioned above guided the development of the water management system. Involvement of the Institute for Sustainable Futures with the 60 L building's water management principles included installing water-saving fixtures and equipment, such as waterless urinals, throughout the facility. Water is collected from roofs and gutters for usage in the home, such as drinking, bathing, and using in sinks. Only the weekly testing of the firefighting system required by law necessitates using the water supply. On the bottom level of the four-story building, two 10,000-L polypropylene storage tanks will hold rainwater collected from the roof. There is constant water quality monitoring, and chemical procedures will be used if necessary. As a result of rainwater treatment, there is a direct link to the scheme's water supply (after it has passed through the system). An automated mechanism activates the link whenever there is a power outage or low tank water. During system testing, the switch can be controlled manually. A future smart water management system should be built on an architecture based on the Internet of Things. Defined here are the fundamental characteristics of a sophisticated and efficient water management system.

An underground tank treats both greywater and blackwater, the waste from toilets and sinks. Sewage treatment alternatives include biological trickle filters and clarifiers, filtration, UV treatment, and chemical modification. Flushing toilets and watering lawns and gardens are two common uses for this purified water. The atrium water feature is intended to consume extra wastewater retrieved, and any further excess will go into the sewer. A connection to the purification system's scheme, water has been put downstream for continuous water quality monitoring, much like the drinking water supply. The use of low-flow resistance cartridge filters, high-efficiency variable speed motors and pumps, and rooftop solar panels are examples of environmentally friendly designs and systems that address the requirement for energy efficiency and reducing carbon dioxide output.

Residential building roofs and other impervious surfaces can collect rainwater in irrigation systems. Rooftop rainwater has a greater quality than rainwater collected from any paved surface. Rooftop water would thus need less treatment than rainfall collected from concrete areas that humans and vehicles often use. In cooling towers, rainwater can reduce blowdown because of its lower total dissolved solids content than scheme water. In addition to toilets, additional indoor sources such as showers and hand basins must be

treated, including screening, oil and grease removal, filtering, and disinfection. The toilet's wastewater, known as blackwater, contains high nutrients.

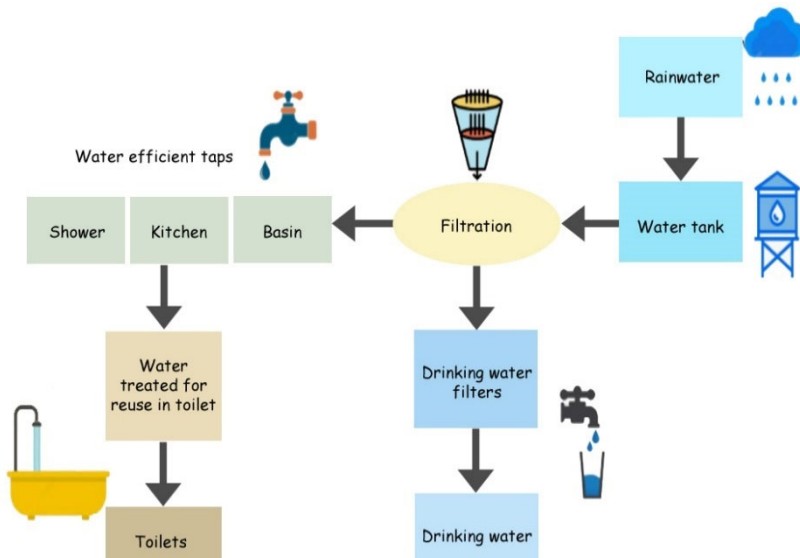

**Figure 5.** Greywater recycling system for the residence.

Disinfection must be done to the highest degree possible when dealing with blackwater due to the high amount of microbiological contamination. There are two primary considerations when matching alternate water sources with suitable end-use. It is possible to ensure that water obtained from a different source is adequate for its intended application using the least amount of treatment possible. The second is the amount of water coming from the alternate source. The ideal match is when the alternative source's supply can fulfill the end consumer's demand at the lowest possible cost. Greywater treatment can be used in various ways, including water for flushing toilets, watering the lawn, and cooling down the air conditioning unit in one's home. For landscape irrigation, treated blackwater is the best option. Sewage that has been micro-filtered and disinfected, for example, can be flushed down the toilet or utilized for irrigation purposes.

Modules for the sensors layer, the gateway, the cloud, and the user interface (UIM). There are typically four core components to an IoT-WQMS (Figure 6). The researchers ultimately turned to IoT implementation. By connecting small sensors to digital computers, networking protocols (such as TCP/IP), and the internet, water quality may be monitored in real-time from anywhere on the planet. Smart water quality monitoring, which relies on the Internet of Things, goes by a few other names. Water quality for drinking, irrigation, aquaculture, municipal waste recycling, and other uses has been tracked with such systems.

Further, these apparatuses check the water standard in lakes, rivers, and the like. Because it leverages current commercially accessible communication infrastructure, the cost of building an IoT-SWM is reduced. The internet is the backbone of IoT-based technologies, so there must be no restrictions on location. The researchers used a local Wi-Fi router and NodeMCU's in-built Wi-Fi functionality to transmit data to a remote cloud server. Fusion-Chart is a program that displays information such as water level, motor on/off status, and tank volume; it communicates with Firebase real-time databases and technologies, including CSS, HTML (Hypertext Markup Language), and JavaScript. To determine the volume of water contained in a spherical tank, the authors used the following Equation (12):

$$X_{water} = \left[\pi r^2\right] Y_{water} \tag{12}$$

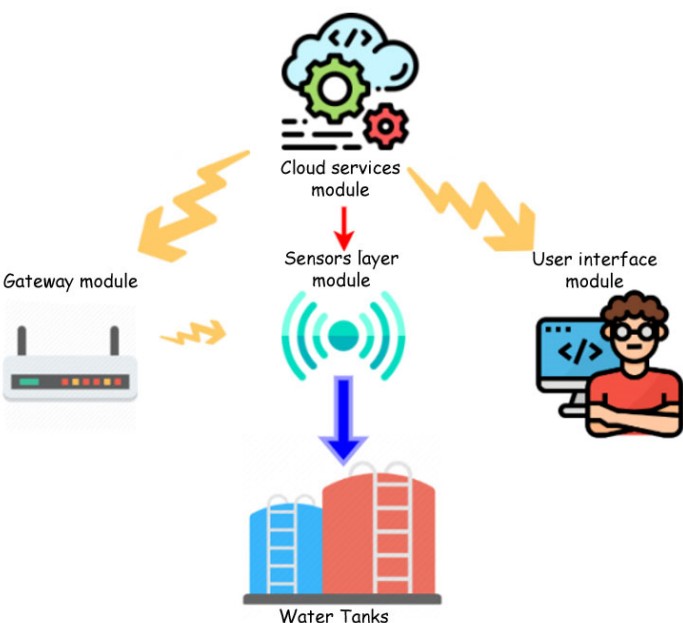

**Figure 6.** Automatic Quality Control for Water.

Equation (12) when $X_{water}$r = $z_{tank}$ $X_{empty}$y. $Y_{water}$, $\pi r^2$, $X_{water}$, $z_{tank}$, and $X_{empty}$ here stand for the volume of water in the tank, the cross-sectional area of the tank, the radius of the tank, the height of the water in the tank at this time, and the total empty volume of the tank. However, modest adjustments would need to be made to the equation to account for various tank designs' unique shapes and sizes. Thanks to this handy website, users may check their water storage tanks from any internet-enabled mobile device. Water leakage was not a factor in the design of this system. The IoT-SWM system has been proposed to develop the stormwater quality, efficiency ratio, water demand ratio, leakage detection ratio, and non-revenue water ratio.

## 4. Numerical Outcome

Internet of Things-based smart water systems can assist prevent these scenarios from occurring and repair the harm that has already been done due to the careless use of water resources. From a freshwater reservoir to the collection and recycling of wastewater, smart water technology makes the entire water supply chain more transparent and controllable. This section includes applications for water management, IoT devices for water management, and smart water treatment methods. Finally, a list of several vital qualities of these systems is framed based on the survey, all of which should be integrated into future designs. As shown in Figure 7, modern IoT-based water quality monitoring solutions have been thoroughly surveyed. All articles focused on meeting the minimum WHO/USEPA standards for potable water, which need systems to be low-cost, portable, low-power, Internet-enabled, real-time, and compliant with these standards. All of these requirements must be met for the water quality index to be considered high. The authors have considered the barest necessities for drinking water and have used machine learning to achieve this goal. The leftover materials have the potential for further development or repurposing, such as in water tank cleaning. The authors offered useful advice for creating a reliable IoT-SWM for residential water. In conclusion, this work is useful for the academic community researching smart water monitoring and the engineering community developing software for smart settings.

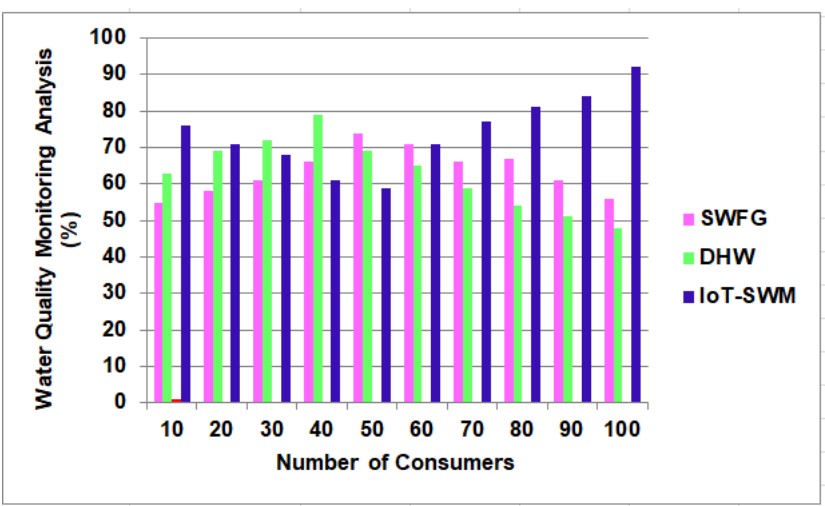

**Figure 7.** Water quality monitoring analysis.

Table 1 shows the analyses of stormwater quality. The quality of rooftop rainwater can be affected by the local environment, roof material, and regular cleaning/maintenance. To further understand the water's diverse properties, we collected and examined 15 rainfall samples from the roofs of different buildings across the area. Samples have been analyzed immediately or within 24 h of being collected. Sterilized glass vials have been used to collect samples, subsequently transported, and kept in a freezer. The study results showed that the water samples reduced levels of chlorine anion and significant cations such as magnesium and calcium. This water meets all the requirements for safe consumption, including pH and electrical conductivity. As a result, the study's findings suggest installing a basic filtration system to catch roof sediments before they reach the storage tank and enhancing stormwater quality analysis by 98.7%.

**Table 1.** Analyses of stormwater quality.

| Number of Consumers | SWFG | DHW | IoT-SWM |
|---|---|---|---|
| 10 | 52 | 65 | 78.5 |
| 20 | 54.5 | 67 | 84 |
| 30 | 55 | 64 | 79 |
| 40 | 58 | 70.8 | 81 |
| 50 | 50.1 | 61.3 | 94 |
| 60 | 56 | 63.7 | 85.7 |
| 70 | 59 | 66 | 97.1 |
| 80 | 57 | 68.4 | 87 |
| 90 | 53 | 69 | 92 |
| 100 | 52.8 | 62 | 98.7 |

Table 2 shows the efficiency ratio. Water efficiency measures the amount of water required for a given operation and the volume of water used or delivered to decrease water waste. Water efficiency differs from conserving water because it focuses on minimizing waste rather than restricting usage, which seems different from water conservation. Planning, producing, distributing, and maintaining the most effective use of water resources are all aspects of water resource management. In the ideal scenario, water resource management planning should include all competing uses and demands for water and be allocated equally to meet all needs. In light of these important factors, an IoT-SWM-based smart water management system design has been given. This design uses machine learning-based predictions to increase the smart management system's efficacy further. By comparison, the suggested strategy improves efficiency by 95.1%. It is essential to have a water management program or plan to ensure that Legionella and other waterborne pathogens are not allowed

to flourish and spread in the water systems of residential buildings. In addition, it has been proposed as future work to develop an Internet of Things (IoT)-based architecture for a smart water management system that incorporates all these crucial features and uses IoT-based predictions to boost its efficacy.

**Table 2.** Efficiency ratio.

| Number of Consumers | SWFG | DHW | IoT-SWM |
|---|---|---|---|
| 10 | 58 | 60 | 75 |
| 20 | 45 | 73 | 78.9 |
| 30 | 52.4 | 66 | 90 |
| 40 | 60.8 | 75.9 | 87 |
| 50 | 59 | 77 | 92 |
| 60 | 65 | 69 | 88 |
| 70 | 55 | 76 | 91 |
| 80 | 50.9 | 71 | 87 |
| 90 | 52 | 77 | 80 |
| 100 | 56 | 60.8 | 95.1 |

Table 3 shows the analysis of the water demand ratio. Demand is measured by the amount of water needed to meet the demands of the consumer population. Due to climate change, freshwater resources are expected to decline in many parts of SA. Human population growth, and changes in land usage and energy production, are other influences. The water's dissolved elements were examined using chemical methods, including the number of suspended particles and the pH value. Population expansion is the primary driver of water demand growth. An increase in water required accompanies a rise in economic prosperity. Smart cities and campuses need a water management system. There has been a notable uptick in the deployment of IoT devices for use in water management. The issues with assessing water quality have been resolved by the availability of cheap sensors linked to IoT devices. This article thoroughly analyzes the current state of smart water management systems and describes the core elements of IoT–based water management platforms. This is due to various factors, including increased agricultural and industrial use, residential use, and concealed virtual water. The suggested approach enhances water demand analysis by 93.6 % compared to the previous methods.

**Table 3.** Analyzing water demand ratio.

| Number of Consumers | SWFG | DHW | IoT-SWM |
|---|---|---|---|
| 10 | 55 | 70 | 74.5 |
| 20 | 43.5 | 63 | 78 |
| 30 | 59 | 76 | 84 |
| 40 | 60 | 65 | 89 |
| 50 | 52 | 78 | 79 |
| 60 | 55 | 69 | 85 |
| 70 | 58.9 | 77 | 91 |
| 80 | 51 | 71 | 94 |
| 90 | 59 | 73.8 | 87 |
| 100 | 53 | 65 | 93.6 |

Table 4 shows the leakage detection ratio. A leak detection device that uses acoustics distinguishes between the noises of leaks and the usual flow of water through the distribution system to find them. Sound waves are used to place a tiny flex on the pipe walls to

determine the real strength of the wall. Detection of a leak in a system is done using leak detection methods. In a wide variety of applications, the procedures are used to seal tanks that contain some substance. Several detecting techniques can be characterized depending on whether the LDC is positioned inside or outside. Real-time water monitoring is an essential feature of any intelligent water management system. Water leaks and overflows can be spotted in real time with the help of monitoring equipment. Keeping up with real-time data demands a constant data connection and a lot of juice. The Internet of Things has drastically altered how we do research and make predictions. Internet of Things (IoT) can be integrated into a water management system to forecast the amount of water needed by a smart house or campus at various times of the day and throughout the year. The same method can be used to meet the water needs of the campus's numerous structures. Similarly, studying and anticipating how things like a wet season may affect water quality is possible.

**Table 4.** Leakage detection ratio.

| Number of Consumers | SWFG | DHW | IoT-SWM |
| --- | --- | --- | --- |
| 10 | 46.3 | 65.9 | 76.8 |
| 20 | 55 | 71 | 87 |
| 30 | 47 | 73 | 82.6 |
| 40 | 49.9 | 65 | 90.2 |
| 50 | 53 | 60.9 | 79 |
| 60 | 57 | 64 | 86 |
| 70 | 45.1 | 72 | 94 |
| 80 | 51 | 71 | 84 |
| 90 | 58 | 78 | 86.9 |
| 100 | 60 | 79.7 | 97.5 |

Further, cloud computing may be used to make decisions at the moment. The system's low energy use is crucial in light of the growing concern over the impact of human energy consumption on the natural world. Using water leakage as an indicator, renewable energy sources may be employed to control water loss. Eventually, the corrosion of a metal container degrades to the point where the material inside the container escapes when leaks occur. IoT leak detection can be an alternative method when other corrosion testing methods fail.

Table 5 shows the analysis of the non-revenue water ratio. Non-revenue water (NRW) has been generated and is lost before reaching the end consumer. Losses can be both actual and perceived (via leaks, sometimes referred to as physical losses). When pipes leak or break, NRW can result in inadequate operations and maintenance, lack of leakage management, and poor quality of subsurface infrastructure. NRW can originate from business losses due to under-registration of client meters, data processing problems, illicit connections, and theft. Unbillable authorized consumption, such as water used by utilities for operations, water used in firefighting, and water provided for free to specific consumer groups, is another reason for NRW. Developing countries' public water utilities can benefit greatly from reducing their NRW.

The water level in the tank or reservoir was evaluated using water level monitoring equipment. Many factors, including pH, total dissolved solids (TDS), dissolved oxygen (DO), and others, were measured by water quality monitoring systems. All methods for gauging water quality looked at the pH value as an indicator of quality. Besides pH and dissolved oxygen, temperature, conductivity, and turbidity are frequently measured while checking water quality. There used to be problems with assessing water quality, but now we have cheap sensors that can be linked to IoT devices; thus, the problem is solved. All current smart water management systems were surveyed, and the key elements of the Internet of Things (IoT)-based water management were identified. Important characteristics such as water level, pH, clarity, salinity, etc., were selected for measurement, and all current

systems were evaluated. The model for calculating the water loss due to leaks was trained and validated using the IoT-SWM approach. The fastest IoT-SWM produced maximum non-revenue water.

**Table 5.** Analyzing non-revenue water ratio.

| Number of Consumers | SWFG | DHW | IoT-SWM |
|---|---|---|---|
| 10 | 61 | 76 | 89 |
| 20 | 75 | 82 | 84 |
| 30 | 65 | 88 | 90.4 |
| 40 | 73 | 79 | 88 |
| 50 | 64 | 75 | 82 |
| 60 | 70 | 83 | 95 |
| 70 | 68 | 78 | 89 |
| 80 | 60 | 86 | 90 |
| 90 | 71 | 89.4 | 91 |
| 100 | 54 | 81 | 98.4 |

## 5. Conclusions

The water sector has been grappling with creating an efficient and long-lasting water system. It is included in the IoT-SWM. People intend to broadcast more data to the cloud and analyze it further to construct some algorithm to determine the tank's lifespan and the proper aspects of leaking. Procedures and actions are determined depending on the threshold, capital cost, and the accessibility of equipment and materials. Even though statistically minimal water savings can be achieved using in-line flow restrictors, they can be much more cost-effective than water-efficient taps in certain situations. If they have been installed as part of normal maintenance visits, the expenses would be lower. They are a low-cost alternative to outdated toilets and are unlikely to save a lot of water. When it the time comes to renovate restrooms, installing water-saving toilets should be considered. To better understand the workings of a crisis-stricken metropolis, this urban crisis feedback analysis tool should be used. With this approach, municipal stakeholders affected by a natural disaster can better plan for a future occurrence of a comparable hazard. We have compiled a list of the most important features of advanced water management systems. There are still barriers to real-time measurement that need minimal energy use. With this in mind, we propose as future work an IoT-based design for a smart water management system that takes into account all of these crucial characteristics and makes use of IoT-based predictions to boost the smart management system's efficacy. As a bonus, future research can use the Internet of Things coverage factor while calculating measurement uncertainty. The authors offer recommendations for the next steps and research groups to join to improve IoT security, lessen the impact of organisms, implement AI/ML approaches, and reduce the entire system's cost. The numerical outcome of the proposed method increases the stormwater quality (98.7%), the efficiency ratio (95.1%), water demand ratio (93.6%), the leakage detection ratio (97.5%), and non-revenue water ratio (98.4%).

**Author Contributions:** Conceptualization, R.A. and S.K.S.; methodology, R.A. and S.K.S.; validation R.A. and S.K.S.; formal analysis, R.A. and S.K.S.; investigation, R.A. and S.K.S.; resources, S.K.S.; data curation, R.A. and S.K.S.; writing—original draft preparation, S.K.S.; writing—review and editing, R.A.; visualization, R.A. and S.K.S.; supervision, R.A. and S.K.S.; project administration, R.A. and S.K.S.; funding acquisition, R.A. All authors have read and agreed to the published version of the manuscript.

**Funding:** This research work was funded by Institutional Fund Projects under grant no. (IFPIP:686-611-1443). The authors gratefully acknowledge technical and financial support provided by the Ministry of Education and King Abdulaziz University, DSR, Jeddah, Saudi Arabia.

**Institutional Review Board Statement:** Not applicable.

**Informed Consent Statement:** Not applicable.

**Data Availability Statement:** Not applicable.

**Conflicts of Interest:** The authors declare no conflict of interest.

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
