# Peer review of "IoT-Based Smart Water Management Systems for Residential Buildings in Saudi Arabia"

_processes, doi:10.3390/pr10112462_

Round 1

Reviewer 1 Report

 In  Saudi Arabia, water management is decisive for both the citizens and the government. The issue is to provide a constant high-quality, low-cost water supply. The presented work presents an Internet of Things-based smart water management (IoT-SWM) system that can be employed in buildings where a continuous supply of water is not accessible; instead, water is stored in large tanks underneath. The GSM module collects water use data from each home in a community and transmits it to the cloud.  The authors suggested a system as a replacement for Saudi Arabia's mechanically operated system.

In this work, procedures and actions are determined depending on the threshold, capital cost, and the accessibility of equipment and materials. Even though statistically minimal water savings can be achieved using in-line flow restrictors, they can be much more cost-effective than water-efficient taps in certain situations. If they have been installed as part of normal maintenance visits, the expenses would be lower. They are a low-cost alternative to outdated toilets, and they are unlikely to save a lot of water. The numerical outcome of the proposed method increases the stormwater quality (98.7%), efficiency ratio (95.1%), water demand ratio (93.6%), leakage detection ratio(97.5%), and non-revenue water ratio (98.4%). 

In my point of view, this study is quite good to publish in the Processes journal,  Section Process Control, and Supervision Special Issue Analysis, Design, and Industrial Application of Intelligent Control Systems.

minor comments

all equations need references (1-11)

please put tables and figures after statements on text not before! 

The English language needs polishing 

Author Response

All equations were referenced. 

Tables and figures were placed after their statements on text. 

The English language was revised and edited.  

Reviewer 2 Report

The authors of the manuscript titled "IoT-based smart water management systems for residential buildings in Saudi Arabia", have given an excellent solution to mitigate the current water problem in Saudi  Arabia. After going through the work, I have the following comments:

  1. Research surveys could be broadened and increased.
  2. Presentation and visualizations should be more engaging and informative.
  3. The efficiency of the proposed method is excellent, but the authors do not validate the proposed method.
  4. A study and presentation of protocols and interfaces used in data transmission and receiving are missing on the cloud side for achieving IoT features.
  5. In point 4, the Numerical Outcome respected author shows Tables 1, 2, 3, 4, and 5, but without project setup or any data set, how analyse or generate those tables?
  6. Lack of comparison with appropriate methods.
  7. Essential references are missing and should be added:
  8.  
  1. Fuentes, H., Mauricio, D. Smart water consumption measurement system for houses using IoT and cloud computing. Environ Monit Assess 192, 602 (2020). https://doi.org/10.1007/s10661-020-08535-4
  2. Jan, F.; Min-Allah, N.; Düştegör, D. IoT Based Smart Water Quality Monitoring: Recent Techniques, Trends and Challenges for Domestic Applications. Water 202113, 1729. https://doi.org/10.3390/w13131729
  3. Machado, Michel R. et al. “Smart Water Management System using the Microcontroller ZR16S08 as IoT Solution.” 2019 IEEE 10th Latin American Symposium on Circuits & Systems (LASCAS) (2019): 169-172.

Author Response

The responses to the respected reviewer's comments are illustrated in the attached file. In addition, the track changes version of the manuscript is attached as well. 

Round 2

Reviewer 2 Report

The authors have addressed my comments, and the updated version has been improved. The author mentions the IoT-SWM proposed as an upcoming work to develop an IoT-based architecture for a smart water management system that considers all these crucial characteristics and uses machine learning-based predictions to boost the smart management system's efficacy. Future works can additionally integrate the IoT coverage element while assessing measurement uncertainty. For an excellent performance of IoT-SWM prototypes, they need some preprocessing data on the water level, which show in numerical outcomes. But this is not the main work of this research. Please check the following research article:

Link 1: https://www.mdpi.com/2073-4441/14/3/309

Link 2: https://www.mdpi.com/1999-5903/12/7/114

Link 3: https://www.mdpi.com/2073-4441/13/13/1729

I suggest the respected author design this research work as a review work based on proposed IoT-SWM prototypes. Also, the authors should need to check the abstract and final discussions again carefully.

Author Response

Addressing the respected reviewer's comments are illustrated in the attached file. The content illustrated in this file is incorporated in the revised version of the paper.   

Round 3

Reviewer 2 Report

The authors have addressed my comments, and the updated version has been improved.